# Regulatory Non-Coding RNAs during Porcine Viral Infections: Potential Targets for Antiviral Therapy

**DOI:** 10.3390/v16010118

**Published:** 2024-01-13

**Authors:** Feng Li, Hao Yu, Aosi Qi, Tianyi Zhang, Yuran Huo, Qiuse Tu, Chunyun Qi, Heyong Wu, Xi Wang, Jian Zhou, Lanxin Hu, Hongsheng Ouyang, Daxin Pang, Zicong Xie

**Affiliations:** 1Key Laboratory of Zoonosis Research, Ministry of Education, College of Animal Sciences, Jilin University, Changchun 130062, China; lifeng21@mails.jlu.edu.cn (F.L.); yu_hao@jlu.edu.cn (H.Y.); qias23@mails.jlu.edu.cn (A.Q.); zhangtianyi23@mails.jlu.edu.cn (T.Z.); huoyr23@mails.jlu.edu.cn (Y.H.); tuqs23@mails.jlu.edu.cn (Q.T.); qicy21@mails.jlu.edu.cn (C.Q.); heyong21@mails.jlu.edu.cn (H.W.); wxi22@mails.jlu.edu.cn (X.W.); zhoujian22@mails.jlu.edu.cn (J.Z.); hulx20@mails.jlu.edu.cn (L.H.); ouyh@jlu.edu.cn (H.O.); 2Chongqing Research Institute, Jilin University, Chongqing 401120, China; 3Chongqing Jitang Biotechnology Research Institute Co., Ltd., Chongqing 401120, China

**Keywords:** pig, ncRNA, viral infections, antiviral therapy

## Abstract

Pigs play important roles in agriculture and bio-medicine; however, porcine viral infections have caused huge losses to the pig industry and severely affected the animal welfare and social public safety. During viral infections, many non-coding RNAs are induced or repressed by viruses and regulate viral infection. Many viruses have, therefore, developed a number of mechanisms that use ncRNAs to evade the host immune system. Understanding how ncRNAs regulate host immunity during porcine viral infections is critical for the development of antiviral therapies. In this review, we provide a summary of the classification, production and function of ncRNAs involved in regulating porcine viral infections. Additionally, we outline pathways and modes of action by which ncRNAs regulate viral infections and highlight the therapeutic potential of artificial microRNA. Our hope is that this information will aid in the development of antiviral therapies based on ncRNAs for the pig industry.

## 1. Introduction

In early research, non-coding RNAs (ncRNA) were often considered “junk regions” outside the coding regions of genes, and researchers focused more attention on the role of coding RNAs than non-coding RNAs. However, recent studies have reported that non-coding RNAs play a similar role in all aspects of life activity [1,2,3]. Many microRNAs are significantly expressed in cancer cells and play a vital role in regulating the progression of cancers. miRNA-155 has been reported as an oncogene in colon, breast, lung, gastric, and hepatocellular carcinomas [4]. miR-215 is markedly up-regulated in glioblastomas [5] and boosts tumor cell growth [6]. In addition, numerous miRNAs have been identified as tumor suppressors; for instance, the miRNA let-7 family has been shown to effectively hinder the progression of various tumors by targeting and suppressing oncogenes, notably c-myc and E2 promoter-binding factor-1 (E2F1) [7]. Furthermore, the miRNA let-7 family has been linked to the regulation of several viral infections, including hepatitis B virus (HBV) [8,9], hepatitis C virus (HCV) [10,11,12,13], Epstein Barr virus (EBV) [14], Kaposi’s sarcoma-associated herpesvirus (KSHV) [15,16], and human papillomavirus (HPV) [17,18]. This highlights the critical roles of ncRNAs in cellular life activities.

Numerous porcine viral infections, including those caused by classical swine fever virus (CSFV), porcine epidemic diarrhea virus (PEDV), porcine reproductive and respiratory syndrome virus (PRRSV), Porcine deltacoronavirus (PDCoV), transmissible gastroenteritis virus (TGEV) and pseudorabies virus (PRV), have resulted in substantial losses for the pig industry [19]. Viruses typically infect target cells by interacting with host receptors and invading host cells. For example, SARS-CoV-2 infects target cells by interacting with the host membrane protein ACE2 through its S protein [20]. Host cells sense viral infections through various pattern recognition receptors (PRRs) and initiate innate immune responses [21], especially activation of the IFN response to produce a variety of antiviral proteins such as RSAD2 and Mx1 to fight against viral infections [22]. However, many viruses have evolved various strategies to evade host immunity, such as inhibiting the IFN response by interacting with proteins at various stages of the IFN pathway, regulating the IFN pathway by phosphorylating or ubiquitylating key signal transduction proteins of the IFN pathway such as STING and MAVS, or hijacking host regulators, including transcription factors and regulatory ncRNAs, to regulate the IFN response [23,24,25]. ncRNAs, as a class of key factors regulating immune responses, have also been widely investigated for their roles in regulating porcine viral infections. Understanding current research progress on the ncRNA mechanism in regulating porcine viral infections can help us to utilize ncRNAs better and thus mitigate the significant losses caused by porcine viral infections in the pig farming industry. In this paper, we will review the current research progress on ncRNAs involved in swine virus infection, summarize the ways in which ncRNAs are involved in swine virus infection, and discuss potential therapeutic strategies based on ncRNAs for swine viral infections.

## 2. Regulatory ncRNA Classification, Production and Function

ncRNAs are a class of RNAs that do not code for proteins; they include microRNAs (miRNAs), long non-coding RNAs (lncRNAs), circular RNAs (circRNAs), PIWI-interacting RNAs (piRNAs), ribosomal RNAs (rRNAs), transfer RNAs (tRNAs), small nuclear RNAs (snRNAs), small nucleolar RNAs (snoRNAs) and small interfering RNA (siRNAs) [3]. Among them, regulatory RNAs, mainly including miRNAs, lncRNAs, circRNAs and piRNAs, are classified according to length and shape. In this paper, we will focus on the regulatory roles of miRNAs, lncRNAs and circRNAs in porcine viral infections. Following this, we will present the production mechanism and mode of action of each of these three types of ncRNAs (Figure 1).

### 2.1. miRNA

miRNAs are a type of minute RNA molecule approximately 22 nucleotides in length. These are often transcribed by RNA Pol II and are regulated by several transcription factors, such as p53, E-cadherin transcriptional repressors (ZEB1) and Myogenic differentiation 1 (MYOD1) [26,27]. Moreover, it has been reported that some viral miRNAs can be transcribed by RNA Pol III [28]. After transcription into primary miRNA (pri-miRNA), the nuclear RNase III Drosha cleaves the pri-miRNA with participation from the cofactor DiGeorge Syndrome Critical Region Gene 8 (DGCR8). This process results in the formation of a small hairpin RNA (pre-miRNA) that is approximately 65 nucleotides in length [29]. Subsequently, the pre-miRNA forms a complex with the protein exportin 5 (EXP5) and the GTP-bound nuclear protein (RAN-GTP) and is released from the nucleus into the cytoplasm [30,31]. In the cytoplasm, Dicer (an RNase type III endonuclease) cleaves pre-miRNAs, releasing a small RNA duplex [32,33]. This duplex is then loaded onto Argonaute (AGO) proteins to form the RNA-induced silencing complex (RISC) [34,35]. Typically, RISCs target mRNAs to inhibit the translation of target genes [36].

### 2.2. lncRNA

RNAs longer than 200 nucleotides that do not encode proteins are commonly referred to as lncRNAs [37]. Like miRNAs, most lncRNAs are transcribed by RNA Pol II [38]. However, the subcellular localization of distinct functional lncRNAs varies. Many lncRNAs possess inadequate internal splice signals and have elongated spans amid 3′ splice regions and branch points, which strengthens their capacity to persist in the nucleus [39]. Furthermore, lncRNAs frequently comprise integrated sequence motifs that assemble specific nuclear factors to augment their nuclear localization. A portion of lncRNAs is also dispatched to the cytoplasm, and their handling and export pathways mirror those of mRNAs. Moreover, lengthy transcripts featuring multiple exons and A/U-rich transcripts need the nuclear RNA export factor 1 (NXF1) pathway for nuclear transportation [40]. Subsequently, cytoplasmic lncRNAs are further sorted into ribosomes and mitochondria, or secreted into exosomes to fulfil biological functions [41]. The function of lncRNA involves the regulation of chromatin structure or gene expression in cis, or departure from the transcription site to perform a biological function in trans [42].

Nuclear lncRNAs perform their biological functions primarily in two ways: 1. Recruiting chromatin modifying factors to form chromatin modifying complexes to regulate gene methylation [43]; 2. Recruiting transcription factors to regulate gene transcription [44]. Cytoplasmic lncRNAs perform their biological functions in different manners: 1. Regulating mRNA stability by complementary pairing with their target RNA sequences [45]; 2. Reducing mRNA degradation by binding miRNAs [46].

### 2.3. circRNA

Circular non-coding RNAs, called circRNAs, are produced by a non-canonical splicing mechanism called backsplicing [47]. This process links the downstream splice donor site to the upstream splice acceptor site in a loop [48]. circRNAs mainly perform their biological functions in the following ways: 1. Acting as microRNA (miRNA) sponges or decoys. They absorb miRNAs to protect mRNAs from degradation [49]; 2. Acting as protein scaffolds to mediate the regulation of protein function and the formation of protein complexes [50]; 3. Recruiting proteins to specific sites or subcellular compartments to carry out specific functions [47]; 4. circRNAs with internal ribosomal entry site (IRES) elements and AUG initiation codons can be translated under certain circumstances to form polypeptides with unique functions [51].

## 3. ncRNA during Porcine Virus Infection

Numerous non-coding RNAs have crucial regulatory roles during porcine virus infection. We present a summary of the ncRNAs implicated in the significant types of porcine virus infections (Table 1) and provide further details about the function of these ncRNAs based on virus species.

### 3.1. CSFV

CSFV is a positive-sense single-stranded RNA virus of *Pestivirus* within the *Flaviviridae* family [52]. There is a scarcity of research regarding ncRNAs in CSFV infection. A recent study showed that CSFV infection of porcine umbilical vein endothelial cells (SUVEC) caused a significant decrease in miR-140 expression, indicating the potential importance of miR-140 in CSFV infection. Further research has shown that miR-140 can complement Ras-related protein 25 (Rab25)-3′UTR in order to hinder Rab25 expression and, as a result, facilitate CSFV replication. Therefore, CSFV inhibits miR-140 expression levels to boost its own infectious ability by restoring Rab25 expression [53]. However, this study did not explain how Rab25 expression promotes CSFV infection. Future research is required to establish the close links between the triad of miR-140, Rab25, and CSFV infection. Additionally, more studies are still necessary to investigate the roles of ncRNAs in CSFV infection.

### 3.2. PEDV

PEDV is a large-enveloped RNA virus belonging to the genus *Alphacoronavirus* [54]. PEDV has a single-stranded positive-sense RNA genome of approximately 28 kb in size that encodes four structural proteins, including spike (S), envelope (E), membrane (M), and nucleocapsid (N) protein, and four nonstructural proteins: 1a, 1b, 3a, and 3b [55]. PEDV invades cells and de-envelopes via S protein binding to special receptors [56]. The PEDV genome is then released into the cytoplasm and immediately translated to produce the replicases ppla and pp1ab. Multiple nsps produced by proteolytical polyproteins form a replication and transcription complex (RTC) that mediates genome replication and protein synthesis of PEDV. The newly synthesized genomic RNA binds to N proteins to form the RNP complex and assembles in the ER-Golgi intermediate compartment (ERGIC) outgrowth, and is finally released via vesicle exocytosis and fusion with the plasma membrane [54]. PEDV has spread widely in swine herds over recent years, causing significant losses to the pig industry. Several studies have demonstrated that the expression profile of ncRNAs, including miRNAs [57,58,59], lncRNAs [58] and circRNAs [60], which play various roles in PEDV infection, undergoes significant changes during PEDV infection. Here, we provide an overview of recent advancements in the study of various ncRNAs during PEDV infection.

#### 3.2.1. miR-221-5p

Related studies have demonstrated the effective up-regulation of miR-221-5p expression in PEDV infection. Moreover, it has been found that miR-221-5p could bind to the 3′UTR of PEDV, leading to effective inhibition of PEDV infection. Furthermore, miR-221-5p increased the production of type I IFN during PEDV infection and activated NF-κB signaling to enhance resistance to PEDV. Additionally, miR-221-5p targeted NF-kappa B inhibitor alpha (NFKBIA) and Suppressor of cytokine signaling 1 (SOCS1), which promoted PEDV infection by inhibiting the production of IFN-β. However, miR-221-5p played a reverse role in this promotion. In conclusion, miR-221-5p is activated following PEDV infection and subsequently inhibits it through various pathways, including targeting viral and host genes. It serves as one of the tools that host cells use to combat PEDV infection [61]. It is worth noting that miR-221-5p boosts the IFN and NF-κB pathways, which leads to inhibition of viral infection. Further studies are needed to determine if this mechanism is effective against other viruses.

#### 3.2.2. miR-615

miR-615 was initially found to inhibit cell proliferation in human breast cancer cells by suppressing protein kinase B beta (AKT2) expression [62]. In addition, miR-615-3p targeted Insulin like growth factor 2 (IGF2) to inhibit tumor growth and metastasis in non-small cell lung cancer (NSCLC) [63]. Until recently, the role of miR-615 in viral infection had not been elucidated. It was predicted via high-throughput sequencing of porcine intestinal epithelial cells (IEC) infected with PEDV that miR-615 may be linked to PEDV infection. Further studies show that miR-615 expression effectively promotes PEDV replication by targeting interleukin-1 receptor associated kinase 1 (IRAK1). Previous studies have shown that PEDV infection activates the NF-κB pathway via a TLR-mediated pathway [64], and that NF-κB acts as a transcription factor to activate the transcription of downstream IFN genes against viral infection [65]. miR-615 has been observed to hamper NF-κB pathway activation and inhibit IFN-III expression by targeting IRAK1, thereby promoting PEDV replication in IEC and MARC-145 cells [66]. Thus, these results suggest that targeting miR-615 to enhance activation of the NF-κB pathway and downstream IFNs may be a promising approach for the treatment of PEDV and other coronaviruses.

#### 3.2.3. miRNA-328-3p 

Previous research has shown that PEDV infection enhances the release of cellular exosomes, which in turn affect the progression of PEDV infection [67]. A recent study found that PEDV infection down-regulates the expression of miRNA-328-3p in exosomes, and further studies revealed that miRNA-328-3p could target and inhibit the expression of zonula occludens 3 (ZO-3). Previous studies have shown that tight junction protein occludin is an entry factor for PEDV, and the expression of ZO-3, a member of the tight junction protein occludin family, also significantly enhanced PEDV infection, possibly by increasing PEDV internalization [68,69,70]. The suppression of miRNA-328-3p has been observed to enhance the expression of ZO-3 and facilitate PEDV infection [71]. Nonetheless, the precise mechanism behind PEDV-mediated regulation of miRNA-328-3p expression and the interplay between ZO-3 protein and PEDV in cells is not yet wholly comprehended, highlighting the need for further research to elucidate the role played by miRNA-328-3p in facilitating PEDV infection. 

#### 3.2.4. miR-let-7e and miR-27b

Milk small extracellular vesicles (sEV) have demonstrated their ability to inhibit PEDV replication, and researchers have identified two miRNAs in sEV that have inhibitory effects on PEDV infection [72]. One of the identified miRNAs, miR-let-7e, has the capacity to hinder the replication of PEDV by targeting its N protein. Another miRNA, miR-27b, can curb the expression of host high mobility group box protein 1 (HMGB1) by targeting its 3′UTR. Previous studies have revealed that the expression of HMGB1 can facilitate PEDV infection. Therefore, miR-27b effectively impedes PEDV infection by targeting host HMGB1 [73]. Further investigations have shown that co-expression of miR-let-7e and miR-27b enhances PEDV inhibition ability, indicating miR-let-7e and miR-27b can jointly synergize to inhibit PEDV infection. Although these results may partly explain the effective resistance of milk sEV against PEDV infection, the presence of more miRNAs with the capability of inhibiting PEDV infection in milk sEV needs further investigation.

#### 3.2.5. LncRNA446

In the sequencing of lncRNA-seq in PEDV-infected and healthy piglets, it has been observed by researchers that lncRNA446 significantly increases after PEDV infection [74]. Subsequently, knock-down of lncRNA446 effectively promotes PEDV infection in IPEC-J2 cells and intestinal organoids. Further studies revealed that lncRNA446 directly binds to ALG-2-interacting protein X (Alix) and represses TRIM25-mediated ubiquitination degradation of Alix, which maintains the expression of tight junction (TJ) proteins. As PEDV infection frequently results in impairment of the epithelial barrier’s integrity, along with modifications in cellular TJs and microfilaments [75], lncRNA446’s maintenance of TJ protein expression supports the intestinal barrier’s integrity and hence boosts resistance to PEDV infection. However, the regulatory mechanism involving Alix and TJ in this study remains unclear, and it is required to examine this novel signaling pathway regulating PEDV infection via lncRNA446/Alix in greater detail. 

### 3.3. PRRSV

PRRSV is an enveloped positive-stranded RNA virus that belongs to the order *Nidovirales*, family *Arteriviridae*. The PRRSV genome RNA is approximately 15 kb in length and contains 11 open reading frames (ORFs), of which four membrane-associated glycoproteins, GP2a, GP3, GP4, and GP5, mediate binding to target cell receptors, such as CD163 and CD169, to initiate viral invasion. Following receptor-mediated endocytosis into the cell, the PRRSV genome is translated to produce the replicase polyproteins pp1a-nsp2TF, pp1a-nsp2N, pp1a, and pp1ab, which are then proteolytically cleaved to produce nsps, which are further assembled into the transcription complex. The newly generated genome is packaged into a nuclear capsid and subsequently removed from the nucleus. The newly generated genome is packaged into nuclear capsids and subsequently budded from intracellular membranes, which are then exocytosed and released [76]. PRRSV can suppress the host’s immune defense system, thus helping more pathogens to establish infections, causing complex and severe porcine respiratory diseases as well as reproductive failure in pregnant sows [77]. Next, we review advances in the study of ncRNAs involved in the regulation of PRRSV infection.

#### 3.3.1. miR-23, miR-378 and miR-505

In a study screening for miRNA binding sites of PRRSV-2 strains, researchers discovered that overexpression of miR-23, miR-505 and miR-378 substantially hinders the replication of PRRSV [78]. Further, luciferase and RISC-IP assays confirmed that miR-23, miR-505 and miR-378 bind to the 3′UTR site of PRRSV and restrain viral replication. Mutations in the binding site lifted the inhibitory effect. Further research has unveiled that miR-23 stimulates I-IFN expression in PRRSV infection through the activation of IRF3/IRF7, whereas miR-505 and miR-378 have no significant impact on I-IFN expression. The combination of these factors leads to the suppressive effects of miR-23, miR-505, and miR-378 on PRRSV infection. Interestingly, the researchers discovered that PRRSV infection does not prompt the expression of miR-23 and miR-505. On the contrary, it curbs the expression of miR-23 and miR-505, which is induced by poly (I:C). Hence, there is a need to investigate the mechanism behind PRRSV’s regulation of the expression of these two miRNAs. 

#### 3.3.2. miR-24-3p

Heme oxygenase-1 (HO-1) is the enzyme that limits the rate of heme degradation [79]. HO-1 and its end products have antioxidant, anti-inflammatory, and antiviral properties. There is evidence to suggest that upregulation of HO-1 expression inhibits the replication of various viruses, including hepatitis C virus (HCV), HIV-1, hepatitis B virus (HBV), and influenza virus (IFV) [80,81,82,83]. PRRSV infection significantly down-regulates HO-1 expression [84,85], and HO-1 expression inhibits PRRSV replication [86]. The researchers found that PRRSV infection induces the expression of miR-24-3p, which could target the 3′UTR of HO-1 mRNA and inhibit HO-1 expression, leading to the impairment of HO-1 expression and thus facilitating PRRSV infection in MARC-145 cells and porcine alveolar macrophages (PAMs) [87]. This could be one of the mechanisms employed by PRRSV to evade host immunity.

#### 3.3.3. miR-125b

As a member of the miR-125 family [88], miR-125b has been reported to inhibit PRRSV replication. The inhibitory impact of miR-125b on PRRSV replication is not reliant on the direct targeting of PRRSV genome or IFN pathway activation, but rather on inhibiting NF-κB activation [89]. This aligns with the known role of miR-125b in human macrophages [90]. Previous studies have shown that PRRSV infection activates NF-κB [91,92], which subsequently promotes PRRSV replication. miR-125b has been demonstrated to be effective in inhibiting PRRSV replication by stabilizing the mRNA encoding κB-Ras2, a crucial inhibitor of NF-κB signaling. The target sequence of miR-125b on κB-Ras2 is conserved across various species, indicating the potential involvement of miR-125b in regulatory roles in viral infections of multiple species. 

#### 3.3.4. miR-26a

The miR-26 family consists of miR-26a and miR-26b, with miR-26a being highly conserved among monkeys and pigs. Research indicates that the miR-26 family efficiently impedes PRRSV replication and gene expression in PAM cells, with miR-26a being more effective than miR-26b in hindering viral growth [93,94]. This largely depends on miR-26a upregulating the expression of type I IFN and IFN-stimulated genes MX1 and ISG15 during PRRSV infection, instead of specifically targeting the PRRSV genome. It is noteworthy that even in the absence of PRRSV infection, miR-26a expression activates IFNs. However, further research is required to identify the direct targets of miR-26a and the precise mechanism by which miR-26a enhances IFN expression.

#### 3.3.5. miR-373

miR-373 is a versatile miRNA that features in many tumor types and can have either oncogenic or tumor suppressor effects [95]. Research into viral infections indicates that miR-373 stimulates replication of HBV [96] and HCV [97]. Moreover, a conducted study demonstrated that PRRSV infection could enhance miR-373 expression by regulating Sp1 expression. Meanwhile, miR-373 suppresses the expression of nuclear factor I-A (NFIA), nuclear factor I-B (NFIB), IRAK1, IRAK4, and interferon regulatory factor 1 (IRF1) via targeted regulation, among which NFIA and NFIB have been acknowledged for inducing IFN-β production [98]. Therefore, miR-373 promotes PRRSV replication by down-regulating the expression of IFN-β. Since miR-373 plays an important role in various viral infections, it is urgently necessary to elucidate its role in various porcine viruses. 

#### 3.3.6. miR-22

Previous studies have shown that HO-1 plays a key role in PRRSV infection [86]. It has been discovered that miR-22 targets and inhibits HO-1 expression directly, by binding to HO-1 3′UTRs, consequently negating HO-1’s antiviral impact on PRRSV [99]. During PRRSV infection of MARC-145 cells, miR-22 is induced significantly to achieve immune evasion. Furthermore, an isoform of miR-22, namely Ssc-miR-22-5p, exhibited up-regulation in Tongcheng pigs that were infected with highly pathogenic PRRSV [100]. This up-regulation might be a contributing factor towards their susceptibility. Additionally, whether miR-22 has additional targets during viral infections remains unclear.

#### 3.3.7. miR-10a-5p

A study based on deep sequencing of small RNAs showed that PRRSV infection significantly induces miR-10a-5p expression. Further investigations revealed that miR-10a-5p has the ability to target the 3′UTR of porcine signal recognition particle 14 (SRP14) mRNA directly, while having no impact on its mRNA levels [101]. However, miR-10a-5p represses the expression of the SRP14 protein in PAM cells by inhibiting the translation of SRP14 mRNA. Subsequently, the knockdown of SRP14 substantially hinders the replication of PRRSV. This indicates that miR-10a-5p impedes the infection of PRRSV by suppressing the levels of SRP14 protein. However, further research is needed to uncover the precise mechanism by which miR-10a-5p regulates levels of SRP14 protein and whether complete dependence on SRP14 is required for inhibition of PRRSV infection by miR-10a-5p. 

#### 3.3.8. miR-218

miR-218 was initially found to hinder invasion and metastasis of gastric cancer cells [102] and has since been closely linked to the development and metastasis of various solid tumors such as prostate, lung, and cervical cancers [103,104,105,106]. A recent study showed that miR-218 is also involved in the regulation of PRRSV infection [107]. By directly targeting the suppressor of porcine cytokine signaling 3 (SOCS3), miR-218 enhances I-IFN, effectively inhibiting PRRSV infection without targeting the viral genome. Conversely, PRRSV infection down-regulates miR-218 expression, impeding IFN expression and enhancing PRRSV’s infectivity. Furthermore, miR-218, which is evolutionarily highly conserved, has also been observed to inhibit other porcine coronaviruses such as PEDV and TGEV. This suggests that miR-218 has the potential to act as a broad-spectrum antiviral agent. 

#### 3.3.9. miR-331-3p and miR-210

Previous studies have shown that miR-331-3p targets TNF-α and substantially reduces its expression in vascular smooth muscle cells (VSMCs). This reduction may potentially regulate the formation of intracranial aneurysms (IAs), which are features of a degenerative disease with mild dilatation of the cerebral arteries [108]. Additionally, miR-331-3p and miR-210 are significantly induced during PRRSV infection [109]. Further studies have demonstrated that both miR-331-3p and miR-210 can directly target PRRSV-2 ORF1b to effectively inhibit PRRSV infection. Additionally, miR-331-3p inhibits TNF-α expression, while miR-210 negatively regulates porcine STAT1 expression. TNF-α is a multifunctional cytokine that not only mediates the host immune response to viruses, but has also been implicated in a variety of lung diseases and is involved in the development of lung injury and fibrosis [110]. As a transcription factor, STAT1 is an important component of the JAK/STAT pathway. STAT1 promotes the expression of pro-inflammatory factors and thus plays a key role in lung injury and other inflammatory diseases [111]. Thus, downregulation of TNF-α and STAT1 results in a reduction of both PRRSV infection and lung injury in pigs. These findings identify potential targets for addressing PRRSV infection, as well as treating inflammation and injury in the lungs. 

#### 3.3.10. ssc-miR-27b-3p

Previous research has demonstrated that miR-27b plays a crucial part in fighting off viral and bacterial infections, such as cytomegalovirus and Salmonella, through its regulation of IFN-γ expression [112,113]. Moreover, overexpressing ssc-miR-27b-3p, a member of the miR-27b family, significantly hampers the replication of PRRSV in both MARC-145 cells and PAMs [114]. However, infection with PRRSV reduces the expression of ssc-miR-27b-3p in the early stage, which might facilitate its immune evasion during the early phases of infection. The precise antiviral mechanisms employed by ssc-miR-27b-3p to inhibit PRRSV replication and the methods used by PRRSV to down-regulate ssc-miR-27b-3p expression for immune evasion are yet to be investigated thoroughly. 

#### 3.3.11. miR-c89

Retinoid X receptor (RXR), a nuclear receptor, is involved in various biological processes, including development, cell differentiation, metabolism, and cell death [115]. Knockdown of RXRβ significantly reduced PRRSV infection, suggesting that RXRβ favors PRRSV infection. However, hosts upregulate the expression of miR-c89 after PRRSV infection, which targets and inhibits RXRβ expression, thereby counteracting PRRSV infection [116]. However, the exact pathway by which RXRβ favors PRRSV replication remains unclear. Notably, the more virulent HP-PRRSV attenuated the expression of miR-c89, which may partially explain the reason for the high virulence of HP-PRRSV.

#### 3.3.12. ssc-miR-30d-R_1

A deep sequencing analysis of miRNomes has been conducted in a study investigating the effects of PRRSV strain LS-4 infection on cells before and after infection. The results show that PRRSV infection has a significant down-regulating effect on the expression of ssc-miR-30d_R-1 in PAM and MARC-145 cells [117]. Conversely, overexpression of ssc-miR-30d_R-1 reduces PRRSV replication by inhibiting the TLR4/MyD88 dependent signaling pathway. This is achieved through targeting the 3′-UTR of the TLR4. In viral challenge trials, ssc-miR-30d_R-1-treated SPF piglets demonstrated reduced PRRSV titers and lung pathology, highlighting ssc-miR-30d_R-1’s inhibitory impact against PRRSV infection. 

#### 3.3.13. miRNA let-7 Family

Let-7, a highly conserved class of miRNAs in animals, was initially identified in C. elegans as a significant gene involved in embryonic development [118]. Subsequently, numerous studies have demonstrated its fundamental contribution to immune regulation, including viral infections and cancer development [119,120]. The let-7 family consists of let-7a, let-7c, let-7d, let-7e, let-7f, let-7g, let-7i, and miR-98. A recent study has demonstrated that PRRSV-2 infection leads to a marked change in the expression of the let-7 family, which subsequently targets PRRSV-2 genomic RNAs, effectively inhibiting their replication [121]. Moreover, the expression of ARID3A transcription factor down-regulates let-7a and let-7f expression, thereby significantly promoting PRRSV-2 replication. As a prospective countermeasure against PRRSV infection, piglets were given injections of the let-7 family expression plasmid. The results indicate that the piglets expressing the let-7 family displayed considerable resistance to PRRSV infection. The markedly lower expression of the let-7 family in Pietrain PAMs compared to Meishan, Changbai and Qingping pigs could be a contributing factor for the lower resistance to PRRSV-2 in Pietrain pigs, when compared to the other three breeds. This finding highlights the potential for future breeding of disease-resistant pigs. 

Another study has demonstrated that let-7f-5p suppresses the expression of myosin heavy chain 9 (MYH9) by binding directly to its 3′UTR. MYH9 is known to interact with the C-terminal structural domain of the PRRSV GP5 protein, causing increased cellular susceptibility to PRRSV infection [122]. As a result, let-7f-5p inhibits PRRSV replication in PAM cells [123]. In contrast, PRRSV infection hinders let-7f-5p expression and promotes the expression of MYH9 to aid viral replication. This mechanism provides yet another way in which the let-7 family restricts PRRSV. 

#### 3.3.14. miR-142-5p

Family with sequence similarity 134, member B (FAM134B) was the initial endoplasmic reticulum autophagy receptor to be discovered [124]. Numerous studies have indicated that FAM134B is a significant limiting factor for viral replication and infection, such as by Ebola (EBOV), Dengue (DENV), West Nile (WNV), and Zika (ZIKA) [125,126]. Recent studies have shown that expression of FAM134B inhibits PRRSV infection by activating type I interferon signaling [127]. However, PRRSV infection is facilitated by up-regulation of miR-142-5p, which directly targets FAM134B expression to inhibit ER autophagy [127]. This reveals another mechanism by which PRRSV achieves immune evasion by regulating miRNA expression, and also provides ideas for the treatment of PRRSV infection. 

#### 3.3.15. miR-181

During an miRNA binding site screening study of the long UTR of PRRSV genomic RNA, researchers discovered a potential binding site for miR-181 in the UTR region of PRRSV for the first time. The overexpression of every member of the miR-181 family impeded PRRSV replication, with the most significant inhibitory effect being observed for miR-181c. Subsequent RISC-IP experiments proved the physical binding effect of miR-181 to PRRSV RNA [128]. Another subsequent study by the investigators demonstrated that miR-181 also targets the 3′UTR of CD163 mRNA, the receptor for PRRSV, leading to a decrease in CD163 expression levels [129]. Several factors combined to produce the inhibitory effects of miR-181 on PRRSV infection. The successful inhibition of replication of porcine highly pathogenic PRRSV, achieved through intranasal administration of miR-181c, fully supports the potential of miR-181c as a therapeutic agent for treating PRRSV infection. 

#### 3.3.16. ssc-miR-124a

PRRSV infection is mediated by a variety of cellular receptors [130], including heparan sulfate (HS) [131], vimentin [132], CD151 [133], CD163 [134], sialoadhesin (CD169) [135], DC-SIGN (CD209) [136], and MYH9 [122]. Among these, CD163 acts as an indispensable receptor that enables the internalization of different species of PRRSV. In a recent study, researchers noticed a significant reduction in the expression of ssc-miR-124a in PAM infected with HP-PRRSV and LP-PRRSV [137]. Bioinformatics prediction and Ago2 immunoprecipitation assay verification indicated that ssc-miR-124a directly targets CD163 mRNA, suppressing CD163 mRNA and protein levels [137]. This implies that ssc-miR-124a is implicated in negatively regulating PRRSV infection and that PRRSV infection enhances its own infection by repressing ssc-miR-124a expression. It is noteworthy that miR-124a exhibits broad-spectrum antiviral action towards Japanese encephalitis virus (JEV), influenza A virus (IAV), and respiratory syncytial virus (RSV) [138,139], highlighting its significance. Moreover, researchers ought to investigate the involvement of miR-124a in other porcine-derived viruses.

#### 3.3.17. miR-204

Autophagy is a conserved cellular recycling mechanism in eukaryotic cells [140]. During viral infection, cells can also eliminate viral proteins, nucleic acids and viral particles through autophagy to maintain the health of the body [141,142]. However, certain viruses, such as PRRSV, Grasshopper cryptozoological virus (GCRV), and pseudorabies virus, can evade or hijack autophagy to infect the host [143,144,145]. PRRSV induces incomplete autophagy, which results in the accumulation of autophagosomes and thus provides a replication site for PRRSV [146]. A recent study revealed that miR-204 was expressed at low levels in PRRSV target cells, specifically PAMs, while it was highly expressed in non-permissive cells, which include porcine peritoneal macrophages (PPMs). Overexpression of miR-204 effectively inhibited PRRSV infection [147]. Further research discovered that miR-204 specifically targeted microtubule-associated protein 1 light chain 3B (MAP1LC3B, LC3B), leading to the inhibition of both PRRSV infection and autophagy. This suggests that one of the reasons for the low infectivity of PRRSV in PPM cells may be that high expression of miR-204 blocks the hijacking of cellular autophagy by PRRSV, thereby impairing PRRSV infection. This study will facilitate the future development of innovative antiviral tactics against PRRSV infection using miR-204.

#### 3.3.18. miR-382-5p

In a previous study, miR-382-5p was reported to target and inhibit HSP60 expression [148]. Additionally, a recent study by researchers reported that PRRSV infection increases miR-382-5p expression, subsequently targeting and inhibiting HSP60 expression [149]. Therefore, this effectively inhibits PRRSV infection. It was also demonstrated through subsequent experiments that HSP60 interacts with MAVS, activating IRF3 and NF-κB while inducing type I interferon production. Therefore, the antiviral effect of type I interferon is avoided by PRRSV-induced miR-382-5p, as it inhibits the expression of HSP60. Based on these findings, miR-382-5p was identified as a promising antiviral target. 

#### 3.3.19. miR-30c

The IFN response is a crucial component of the reaction to viral infections. Nonetheless, research has demonstrated that PRRSV infection can evade the immune system by controlling miR-30c expression. miR-30c facilitates a conducive environment for PRRSV infection by suppressing the expression of JAK1 and IFNAR2, and consequently reducing the activation of the IFN-I pathway, which impairs downstream ISGs expression [150,151]. This observation offers new perspectives into PRRSV immune evasion. 

#### 3.3.20. miR-506

During a screening for miRNAs that target CD151, a receptor infected with PRRSV, investigators discovered that miR-506, miR-124, and miR-9 all target CD151 3′UTR [152]. However, further studies indicated that solely miR-506 overexpression effectively diminished both CD151 mRNA and protein expression levels. In addition, PRRSV infection was also considerably impeded in Marc-145 cells that overexpressed miR-506. These findings indicate that miR-506 can successfully impede PRRSV infection by targeting CD151 gene expression. 

### 3.4. PDCoV

PDCoV is an enveloped single-stranded positive-sense RNA virus that belongs to the genus *Deltacoronavirus* in the family *Coronaviridae* of the order *Nidovirales*. The PDCoV genome is approximately 25.4 kb in length and encodes several proteins, including open reading frame 1a/1b (ORF1a/1b), spike (S), envelope (E), membrane (M), non-structural protein 6 (NS6), nucleocapsid (N), NS7, and NS7a [153]. The current study suggested that PDCoV entry into cells may be dependent on S protein interaction with APN [154]. PDCoV infection causes acute watery diarrhea in suckling pigs, which can be severe enough to cause death [155]. There is a scarcity of information about the involvement of host ncRNAs in regulating PDCoV infection. In a recent study, investigators used small RNA and RNA sequencing to screen eight representative miRNAs that are differentially expressed in PDCoV-infected IPEC-J2 cells. The screened miRNAs included miR-124a, miR-148a-5p, miR-30c-3p, miR-374b-3p, miR-194a-3p, miR-499-3p, miR-4332 and miR-1285 [156]. The first two of these microRNAs (miRNAs) have been linked to regulating various viral infections. For instance, miR-124a specifically targets the receptor CD163 to inhibit infection caused by PRRSV [137]. Similarly, miR-148a-5p regulates the duck Tembusu virus by targeting SOCS1 [157]. Moreover, overexpressing ssc-miR-30c-3p or ssc-miR-374b-3p suppresses PDCoV replication in IPEC-J2 cells. However, the mechanism behind this suppression remains unclear. Whether other miRNAs that are differentially expressed after PDCoV infection could be involved in the regulation of PDCoV infection requires further study.

### 3.5. TGEV

TGEV is a single-stranded positive-sense RNA virus that belongs to the family *Coronaviridae* of the order *Nidovirales*. The TGEV genome is approximately 27.6 to 31.5 kb in size and encodes a variety of proteins, including ORF1a, ORF1b, S, ORF3a, ORF3b, E, M, N, and ORF7 [158]. TGEV invades target cells with the help of S protein–APN interaction [159]. TGEV infection causes symptoms of vomiting, diarrhea and dehydration and is almost 100% lethal in piglets [160]. Here, we provide an overview of recent advancements in the study of various ncRNAs during TGEV infection.

#### 3.5.1. miR-30a-5p

The miR-30 family, consisting mostly of five members, namely miR30a to miR30e, has been extensively implicated in the onset of cancer and viral infections [150,161,162,163]. The infection by the Transmissible gastroenteritis virus (TGEV) promotes substantial endoplasmic reticulum (ER) stress and activates all three unfolded protein response (UPR) pathways, indicating the significance of ER stress in TGEV infection [164,165,166]. A study has shown that activated IRE1α-mediated UPR reduces miR-30a-5p expression during TGEV-induced prolonged endoplasmic reticulum stress [167]. Additional research indicates that miR-30a-5p enhances the IFN-induced antiviral signaling pathway by directly targeting SOCS1 and SOCS3, negative regulators of IFN signaling, thereby preventing TGEV infection. However, TGEV infection inhibits miR-30a-5p expression in vivo and notably up-regulates the expression of SOCS1 and SOCS3 in the ileum. This action allows TGEV to inhibit the antiviral signaling pathway induced by IFN, thereby enabling immune evasion. In addition, induction of SOCS1 and SOCS3 to counteract the IFN antiviral response is also observed in other viral infections, such as SARS-CoV [168], herpes simplex virus 1 [169], respiratory syncytial virus [170], HCV [171], and IAV [172], which implies that miR-30a-5p may have a wide-spectrum antiviral capacity and deserves further investigation in the future.

#### 3.5.2. miR-27b

Previous research has demonstrated that TGEV infection leads to apoptosis in PK15 cells [173] and that RUNX1 has a crucial function in regulating the process [174]. A study showed that miR-27b could reduce the expression of RUNX1 by targeting its 3′UTR [175], thereby attenuating the activation of BAX by RUNX1, which in turn attenuated apoptosis by regulating the mitochondrial pathway. By contrast, miR-27b expression was significantly reduced during TGEV infection [176], indicating that miR-27b may be employed by TGEV to regulate apoptosis in PK15 cells. Another study discovered that miR-27b-3p could impede TGEV replication by targeting suppressor of porcine cytokine signaling 6 (SOCS6) directly [177]. However, during porcine epidemic diarrhea virus (PEDV) infection, PEDV-induced endoplasmic reticulum (ER) stress activates myo-inositol-requiring enzyme 1 (IRE1), which encodes a transcription factor, X-box binding protein 1 (Xbp1s). This transcription factor effectively inhibits the production of miR-27, reducing the production of miR-27b-3p and ultimately leading to the reversal of miR-27b-3p-mediated anti-PEDV activity. The findings indicate that PEDV manipulates host ER stress sensor IRE1 to suppress host miR-27b-3p expression, thus improving viral survival. Significantly, the study revealed that overexpression of miR-27b suppresses the replication of various coronaviruses, including PEDV, TGEV, and PoRV, indicating that miR-27b has wide antiviral activity. However, the exact mechanism of miR-27b and its resistance to diverse porcine-derived viruses requires exploration in forthcoming studies.

#### 3.5.3. miR-4331

The genome of TGEV features a leading sequence positioned at the 5′ end and a poly (A) tail located at the 3′ end. TGEV genes are arranged in a 5′-Rep-S-3a-3b-E-M-N-7-3′ sequence, where gene 7 is positioned at the 3′ end of the genome [178]. Research indicates that TGEV infection significantly induces miR-4331, and overexpression of miR-4331 leads to the transcriptional repression of TGEV gene 7 [176]. Nevertheless, there is no target for miR-4331 in TGEV gene 7. Further research indicates that miR-4331 impacts the transcription of TGEV gene 7 by targeting and inhibiting the expression of intracellular CDCA7, an important transcriptional regulator that is involved in the control of the cell cycle and the action of oncoproteins [179]. The specific mechanism by which CDCA7 influences the transcription of TGEV gene 7 requires further investigation.

Furthermore, besides the aforementioned ncRNAs, TGEV infection notably regulates other ncRNAs, such as lncRNA TCONS_00058367 and ssc_circ_009380, but the precise mechanisms underlying the roles of these ncRNAs in the regulation of TGEV infection remain unknown. lncRNA TCONS_00058367 is significantly down-regulated and may play an important regulatory role in TGEV infection-induced inflammatory responses by negatively regulating the phosphorylation of the transcription factor p65 (p-p65) in TGEV-infected IPEC-J2 cells, thereby decreasing its antisense gene promyelocytic leukemia (PML) [180]. ssc_circ_009380, on the other hand, has been reported to mediate TGEV-induced inflammatory response through adsorption of miR-22. When miR-22 is overexpressed, p-IκB-α is reduced in TGEV-infected IPEC-J2 cells, leading to the accumulation of p65 in the nucleus, whereas silencing of ssc_circ_009380 inhibits the accumulation of p65 and phosphorylation of IκB-α in the nucleus [181]. Further experiments demonstrated that ssc_circ_009380 attenuates TGEV-induced activation of the NF-κB pathway by directly binding miR-22. There are few reports on the involvement of ncRNAs in regulating TGEV infection, and more studies are expected in the future.

### 3.6. PRV

PRV is an enveloped, double-stranded linear DNA virus belonging to the subfamily *Alphaherpesvirinae* within the family *Herpesviridae*. PRV relies on the interaction of gC protein with the acetylheparin sulphate proteoglycan in the extracellular matrix to attach to the cell, and then PRV gD protein binds to specific cellular receptors and mediates the fusion of the viral envelope and the cytoplasmic membrane and thus entry into the target cell with the help of PRV gB, gH and gL [182,183]. Next, we review the latest research on ncRNAs involved in PRV infection.

#### 3.6.1. LNC_000641

A recent study showed that PRV infection significantly increases the expression of LNC_000641 in 3D4/21, PK15 and ST cells [184]. Subsequent studies showed that overexpression of LNC_000641 inhibits both IFN-α production and JAK and STAT1 phosphorylation, impairing downstream ISG transcript levels. This results in LNC_000641 expression favoring PRV infection, which may serve as a potential means for PRV to evade the host IFN response. The mechanism by which LNC_000641 regulates modifications in JAK/STAT1 and IFN is unclear and warrants investigation in the future.

#### 3.6.2. lncA02830

In a lncRNA library screen of gE-TK-PRV-II infected cells, researchers found that PRV infection significantly down-regulates the expression of lncA02830 [185]. Overexpressing lncA02830 resulted in a significant reduction in mRNA levels of IRF3, IFNβ, and MX1. Conversely, when silencing lncA02830, the mRNA levels of IRF3, IFNβ, and MX1 were significantly increased, effectively inhibiting PRV infection. This implies that the lncA02830 gene functions as a critical regulator during PRV infection, and suggests that PRV might modulate the immune response of the host by exploiting lncA02830 to facilitate self-infection. Nonetheless, it remains unclear whether lncA02830 directly modulates the host’s IFN response. The precise mechanism by which lncA02830 exerts its antiviral activity against PRV requires further investigation.

**Table 1 viruses-16-00118-t001:** ncRNAs involved in the regulation of porcine viral infections.

Virus	ncRNA (Name)	Regulation	Targets	Effect on Virus Infection	Reference
CSFV	miR-140	down	Rab25	inhibit	[53]
PEDV	miR-221-5p	up	PEDV 3′UTR; nFKBIA; SOCS1	inhibit	[61]
	miR-615	down	IRAK1	promote	[66]
	miRNA-328-3p	down	ZO-3	inhibit	[71]
	miR-let-7e	up	PEDV N protein	inhibit	[72]
	miR-27b	up	HMGB1 3′UTR	inhibit	[72]
	lncRNA446	up	Alix	inhibit	[75]
PRRSV	miR-23	unchanged	PRRSV 3′UTR	inhibit	[78]
	miR-378	unchanged	PRRSV 3′UTR	inhibit	[78]
	miR-505	unchanged	PRRSV 3′UTR	inhibit	[78]
	miR-24-3p	up	HO-1 3′UTR	promote	[87]
	miR-125b	down	κB-Ras2	inhibit	[89]
	miR-26a	up	unknown	inhibit	[93,94]
	miR-373	up	NFIA, NFIB, IRAK1, IRAK4, and IRF1	promote	[98]
	miR-22	up	HO-1 3′UTR	promote	[99]
	miR-10a-5p	up	SRP14 3′UTR	inhibit	[101]
	miR-218	down	SOCS3	inhibit	[107]
	miR-331-3p	up	PRRSV ORF1b; STAT1	inhibit	[109]
	miR-210	up	PRRSV ORF1b	inhibit	[109]
	ssc-miR-27b-3p	down	unknown	inhibit	[114]
	miR-c89	up	RXRβ	inhibit	[116]
	ssc-miR-30d-R_1	down	TLR4 3′UTR	inhibit	[117]
	miRNA let-7 family	difference	PRRSV genomic RNAs; MYH9	inhibit	[121]
	miR-142-5p	up	FAM134B	inhibit	[127]
	miR-181	unchanged	PRRSV RNA; CD163 3′UTR	inhibit	[129]
	ssc-miR-124a	down	CD163 mRNA	inhibit	[137]
	miR-382-5p	up	HSP60	promote	[149]
	miR-30c	up	JAK; IFNAR2	promote	[150,151]
	miR-506	unknown	CD151 3′UTR	inhibit	[152]
PDCoV	ssc-miR-30c-3p	up	unknown	inhibit	[156]
	ssc-miR-374b-3p	up	unknown	inhibit	[156]
TGEV	miR-30a-5p	down	SOCS1; SOCS3	inhibit	[167]
	miR-27b	down	SOCS6	inhibit	[73]
	miR-4331	up	CDCA7	inhibit	[176]
	ssc_circ_009380	down	miR-22	undefined	[181]
	lncRNA TCONS_00058367	down	p-p65	undefined	[180]
PRV	lnc_000641	up	unknown	promote	[184]
	lncA02830	down	unknown	promote	[185]

## 4. The Ways ncRNAs Regulate Viral Infection

Host-encoded ncRNAs frequently participate in the regulation of viral infections via various mechanisms. Simultaneously, numerous viruses have developed tactics to manipulate host ncRNAs to evade immunity by targeting immune pathways. It is crucial to comprehend how ncRNAs in porcine viruses regulate viral infection and which pathways are involved to create ncRNA-based antiviral therapeutics. There are four primary approaches by which ncRNAs regulate porcine virus infection: 1. Directly targeting the viral genome; 2. Targeting viral receptors; 3. Regulating the TLR, NF-κB or IFN pathway; 4. Targeting other host factors. We will now elaborate on the specific mechanisms involved in these approaches (Figure 2).

### 4.1. Targeting Viral Genomes

Direct targeting of the viral genome by miRNAs to mediate inhibition of viral infection is considered to be the most effective and direct antiviral strategy for the host. To date, a number of miRNAs have been reported to target the genomes of porcine viruses. For example, during porcine epidemic diarrhea virus (PEDV) infection, miR-221-5p inhibits virus replication by binding to its 3′UTR, consequently suppressing its replication [61]. Similarly, miR-let-7e inhibits PEDV infection by targeting its nucleocapsid (N) protein [72]. Furthermore, it has been reported that miR-23, miR-378, and miR-505 inhibit PRRSV replication by targeting the 3′UTR of PRRSV [78]. Additionally, miR-331-3p and miR-210 have demonstrated inhibitory effects on PRRSV infection by targeting ORF1b of PRRSV [109]. Moreover, the let-7 family of miRNA and miR-181 have been shown to extensively target PRRSV RNA, resulting in significant antiviral activity [129].

### 4.2. Targeting Viral Receptors

Receptor expression and integrity are necessary initial steps in viral infection. Thus, regulating the expression of targeted receptors by miRNA is deemed an effective antiviral strategy. CD163 is targeted by both miR181 and ssc-miR-124a, resulting in a successful inhibition of PRRSV infection [129]. Moreover, CD151, another PRRSV receptor, is targeted by miR-506, leading to an effective inhibition of PRRSV infection [152]. In addition, miR-124, and miR-9 are also found to target CD151, and their effects on PRRSV infection need to be further investigated in the future [137].

### 4.3. Regulating TLR, NF-κB and IFN Pathways

The IFN, TLR and NF-κB pathways play a crucial role in the immune response of cells to viral infection. Additionally, many ncRNAs regulate viral infection by modulating these pathways. ssc-miR-30d_R-1 inhibits the TLR4/MyD88-dependent signaling pathway by targeting and suppressing TLR4 expression, which leads to the blockage of PRRSV infection [117]. On the other hand, miR-615 promotes PEDV proliferation by targeting IRAK1 to inhibit the activation of the NF-κB pathway and IFN production [66]. miR-125b inhibits PRRSV infection by targeting and suppressing IkB expression of kB-Ras2, a negative regulator of IκB, thus enhancing the NF-κB pathway [89]. miR-373 and lncRNA TCONS_00058367 regulate NF-κB pathway activation by modulating p65 [98,180]. MiR-382-5p inhibits HSP60, reducing IRF3 activation, which ultimately promotes PRRSV infection [149]. For the interferon pathway, miR-30c and miR-331-3p targeted IFNAR2 and STAT, which are key factors of the interferon pathway, and impeded the pathway’s signaling, leading to greater susceptibility to viral infections [150,151]. On the contrary, miR-221-5p [61], miR-30a-5p [167], miR-218 [107], and miR-27b [73] enhanced the activation of the interferon pathway by targeting and inhibiting SOCS1, 3, and 6, which are crucial negative regulators of the interferon pathway, thereby boosting the host’s antiviral capability. 

### 4.4. Targeting Other Cellular Factors

In addition to the above approaches, ncRNAs can also achieve regulation of viral infection by targeting other factors in the cell. For example, miR-24-3p regulates viral infection by targeting HO-1, an important anti-PRRSV factor [87]. lncRNA446 affects PEDV infection by targeting Alix to alter cellular tight junctions [75]. Unlike the unified mechanism of ncRNA targeting the IFN pathway to regulate viral infection, these findings imply that the mechanism of ncRNAs in regulating viral infection is more complex and requires deeper studies on different viruses and even different cells.

## 5. amiRNA-Based Antiviral Therapy

Conventional antiviral therapies include the use of vaccines to induce antibody production in the host against viral infections, as well as targeting key viral genes by siRNA to inhibit viral infections. However, many viruses, especially RNA viruses, tend to evade antibody or siRNA targeting through mutation, which poses a great challenge to conventional antiviral therapy. Because ncRNAs play an important role in regulating viral infections, a range of ncRNA-based antiviral therapies have been developed. A widely reported non-coding RNA therapy is artificial microRNA (amiRNA). Compared to conventional antiviral strategies that target viruses via antibodies and use siRNAs to target viral mRNAs, amiRNAs do not require stringent base complementary pairing and are better able to cope with viral mutation escape; hence, amiRNAs typically have lower off-target effects and better inhibitory efficacy. amiRNAs can exert antiviral functions by targeting different types of viruses or host mRNAs. Viral infection could be effectively inhibited by designing amiRNAs that target viral mRNAs (Table 2). The amiRNA-349 and amiRNA-1447 targeting the conserved regions of PEDV N and S genes, respectively, effectively inhibited PEDV infection [186]. amirGP5-370 and amirM-263-M-263 targeting of PRRSV GP5 or M proteins, respectively, effectively inhibited PRRSV infection [187]. Another strategy is to design amiRNAs that target viral receptors. Sialoadhesin (Sn) and CD163 are two important receptors for PRRSV, and PRRSV infection can be effectively inhibited by designing amiRNAs that target Sn and CD163 [188]. However, how amiRNAs can achieve more efficient delivery still needs further study.

## 6. Perspectives

As a part of host immunity, ncRNAs play an important role in combating porcine viral infections in a variety of ways, including targeting the viral genome and host-related factors and enhancing the host IFN pathway. However, many viruses have evolved immune evasion mechanisms that hijack ncRNAs to suppress host immunity. Understanding the ncRNAs involved in the regulation of porcine virus infection and the mechanisms that regulate viral infection is crucial for us to develop better antiviral therapies. However, we still lack an in-depth understanding of ncRNAs in porcine-origin viral infections, which may be due to a lack of means to study ncRNAs. The outbreak of SARS-CoV-2 has added a number of new research ideas to virology research, such as genome-wide library screening mediated by CRISPR. However, current library screening is mainly focused on the study of coding genes. Library screening for non-coding genes may be a new direction for studying viral infections in the future, and may help us to deepen our understanding of the mode of action of ncRNAs in viral infections from a global perspective. In addition, the application of ncRNA therapy in porcine viral infections is still lacking, mainly focusing on shRNA or amiRNA-mediated RNAi. The outbreaks of porcine viruses such as ASFV, PEDV, and PPRSV and the difficulties in vaccine development in recent years have highlighted the need to find new avenues for the development of antiviral therapies. ncRNA therapies have demonstrated greater advantages than traditional antiviral therapies, including lower toxicity and off-targeting, and greater ability to cope with viral mutations. Further research is needed to develop more effective ncRNA therapeutics that take full advantage of the benefits of ncRNAs in regulating viral infections.

## Figures and Tables

**Figure 1 viruses-16-00118-f001:**
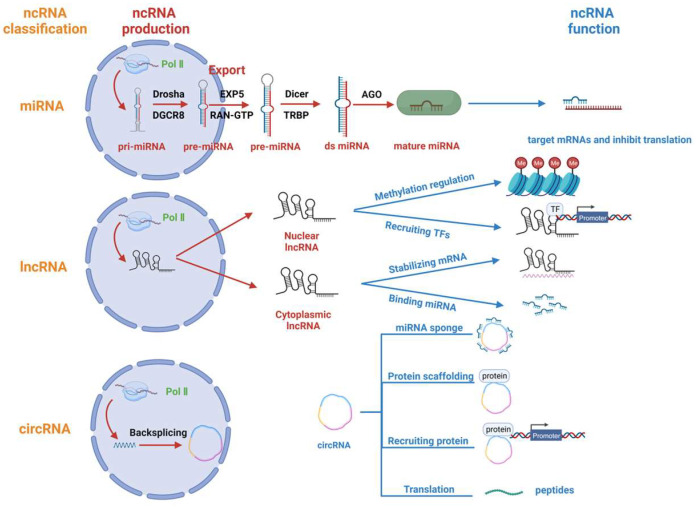
Regulatory ncRNA classification, production and function. Regulatory non-coding RNAs (ncRNAs) comprise miRNAs, lncRNAs, and circRNAs. miRNAs are transcribed from Pol II to form pri-miRNAs, which are processed by Drosha and DGCR8 to form pre-miRNAs. The pre-miRNAs are exported from the nucleus with the help of EXP5 and RAN-GTP and are processed by Dicer. Finally, the double-stranded miRNA (ds miRNA) is loaded onto the AGO protein and functions to inhibit translation. lncRNAs are mainly transcribed by Pol II and are partially exported from the nucleus to the cytoplasm. Nuclear lncRNAs are primarily responsible for regulating gene methylation and recruiting transcription factors, while cytoplasmic lncRNAs primarily bind and stabilize mRNAs and also bind and repress microRNAs. CircRNAs undergo backsplicing after being transcribed by Pol II, and their main functions include acting as miRNA sponges, serving as protein scaffolds, recruiting proteins and partially transcribing them into polypeptides with unique functions.

**Figure 2 viruses-16-00118-f002:**
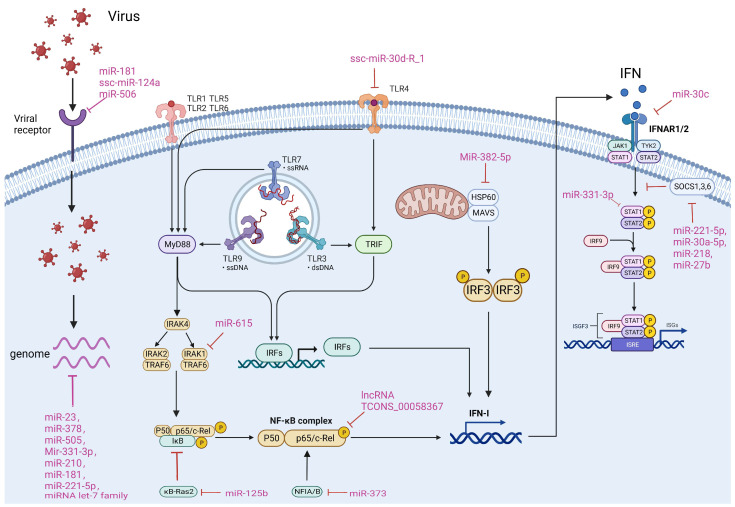
ncRNAs affect viral infection in different ways. The main ways in which ncRNAs regulate viral infection include direct targeting of the viral genome and targeting of viral receptors, as well as through modulation of the host TLR, NF-κB and IFN pathways. The ncRNAs and their targets that are involved in different mechanisms of porcine-derived virus infection are summarized in the Figure.

**Table 2 viruses-16-00118-t002:** amiRNAs against porcine viral infections.

amiRNA	Target	Virus	Reference
amiRNA-349	PEDV N	PEDV	[186]
amiRNA-1447	PEDV S	PEDV	[186]
amiRGP5-370	PRRSV GP5	PRRSV	[187]
amiRM-263-M-263	PRRSV M	PRRSV	[187]
amiRNA-Sn	Sn	PRRSV	[188]
amiRNA-CD163	CD163	PRRSV	[188]
amiR3UTR	PRRSV 3′UTR	PRRSV	[189]

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
