# Peer review of "Regulatory Non-Coding RNAs during Porcine Viral Infections: Potential Targets for Antiviral Therapy"

_viruses, 2024, doi:10.3390/v16010118_

Round 1

Reviewer 1 Report

Comments and Suggestions for Authors

Overall a nicely written review on ncRNAs and their roles in the regulation of pig virus replication and host responses.  This review would be especially helpful to those who have limited knowledge of ncRNA and its function.  Only minor grammar issues.

Comments on the Quality of English Language

Overall, the English is good.  Suggest that the authors review the paper one more time to correct minor issues.

Reviewer 2 Report

Comments and Suggestions for Authors

Comments on the Quality of English Language

Moderate editing of English is required.
